# Phosphodiesterase 1A physically interacts with YTHDF2 and reinforces the progression of non-small cell lung cancer

Chong Zhang[1,2†], Zuoyan Zhang[3,4†], Yueyi Wu[2], Yuchen Wu[5†], Jing Cheng[2,6], Kaizhi Luo[2,7], Zhidi Li[2,6], Manman Zhang[2,6], Jian Wang[1*], Xuesen Zhang[8*], Yangling Li[9*]

[1]Department of Oncology, Shangyu People's Hospital of Shaoxing, Shaoxing, China; [2]Department of Pharmacy, School of Medicine, Hangzhou City University, Hangzhou, China; [3]Department of Pharmacy, The Affiliated Hospital of Northwest University, Shanxi, China; [4]Department of Pharmacy, Ningbo First Hospital, Ningbo, China; [5]Department of Clinical Medicine, The First School of Medicine, Wenzhou Medical University, Wenzhou, China; [6]College of Pharmaceutical Sciences, Zhejiang University, Hangzhou, China; [7]Department of Pharmacy, Zhejiang University of Technology, Hangzhou, China; [8]Hangzhou Lin'an Traditional Chinese Medicine Hospital , Affiliated Hospital, Hangzhou City University, Hangzhou, China; [9]Department of Clinical Pharmacology, Key Laboratory of Clinical Cancer Pharmacology and Toxicology Research of Zhejiang Province, Affiliated Hangzhou First People's Hospital, School of Medicine, Westlake University, Hangzhou, China

*For correspondence:
2538765918@qq.com (JW);
xuesenz@163.com (XZ);
liyangling1215@163.com (YL)

†These authors contributed equally to this work

Competing interest: The authors declare that no competing interests exist.

## eLife Assessment

This manuscript provides **valuable** mechanistic insight into NSCLC progression, both in terms of tumour metastasis and the development of chemoresistance. The authors draw upon a range of techniques and assays and the evidence shown is **solid** and has been strengthened by incorporation of suggestions by the two reviewers. The work presented will be of interest to cancer biologists and more broadly to those interested in NSCLC translational studies.

**Abstract** Non-small cell lung cancer (NSCLC) is the most common subtype of lung cancer, and the prognosis is poor due to distant metastasis. Thus, there is an urgent need to discover novel therapeutic targets and strategies to overcome metastasis. A series of in vitro and in vivo phenotype experiments were performed to investigate the role of phosphodiesterase 1A (PDE1A) in NSCLC. The RNA binding protein immunoprecipitation (RIP) assay, messenger RNA (mRNA) stability assay, and LC-MS/MS were performed to investigate the molecular mechanisms of PDE1A in NSCLC progression. PDE1A has been shown to promote metastasis and epithelial-mesenchymal transition (EMT) progression of NSCLC. In addition, NSCLC cells overexpressing PDE1A promoted angiogenesis by regulating exosome release. IL-6/JAK/STAT3 signaling pathway was highly enriched in PDE1A-coexpressed genes, and PDE1A promoted NSCLC metastasis by activating the STAT3 pathway. GO enrichment analysis of PDE1A-interacting genes showed that PDE1A might interact with YTHDF2 and participate in m6A-containing RNA binding. The binding between PDE1A and YTHDF2 was verified, and PDE1A regulated the STAT3 pathway by interacting with YTHDF2. The mechanism of the YTHDF2/PDE1A complex in regulating the STAT3 pathway was predicted by overlapping YTHDF2-interacting RNAs and genes coexpressed with YTHDF2 and STAT3. The interactions between YTHDF2 and target mRNAs were predicted, and there were three predicted targets of YTHDF2 with

high scores: NRF2, SOCS2, and MET. Indeed, PDE1A interacted with YTHDF2, destabilized SOCS2, and activated the STAT3 pathway. Mechanistic data uncover a novel PDE1A/YTHDF2/STAT3 axis driving NSCLC metastasis and suggest potential therapeutic strategies for metastatic disease.

## Introduction

Lung cancer is one of the most frequently diagnosed cancers and the leading cause of cancer-related mortality worldwide (*Thai et al., 2021*). Non-small cell lung cancer (NSCLC) represents approximately 85% of lung cancers and is the most common subtype of lung cancer (*Alduais et al., 2023*). Despite rapid advances in the clinical treatment of NSCLC in recent years, the prognosis of NSCLC patients is still poor due to recurrence and distant metastasis (*Xia et al., 2021*). However, the mechanism of NSCLC metastasis is still poorly understood.

Phosphodiesterases (PDEs) are a class of enzymes that hydrolyze cyclic adenosine monophosphate (cAMP) and cyclic guanosine monophosphate (cGMP), reducing the signaling of these important intracellular second messengers (*Delhaye and Bardoni, 2021*). PDEs consist of 11 family members, and each family member contains multiple subtypes (*Peng et al., 2018*). PDEs are being pursued as therapeutic targets in multiple diseases, including the cardiovascular system, metabolism, pulmonary system, nervous system, immunity, and cancers (*Baillie et al., 2019*). For example, PDE3 inhibitors are used to treat heart failure and peripheral artery disease, and PDE4 inhibitors are approved to treat inflammatory diseases (*Hsien Lai et al., 2020*). Multiple studies have shown that PDEs, such as PDE4 and PDE5, play a vital role in the progression of tumors and are regarded as potential therapeutic targets for cancer treatment (*Hsien Lai et al., 2020*; *Huang et al., 2020*). The PDE1 family member has three subtypes, PDE1A, PDE1B, and PDE1C, with different affinities for cAMP and cGMP (*Samidurai et al., 2021*). The effect and mechanism of PDE1 in regulating cancer progression remain elusive.

$N^6$-methyladenosine (m$^6$A) is the most abundant RNA modification, and the process of m$^6$A modification is reversible: m$^6$A is installed by 'writers', removed by 'erasers', and recognized by 'readers' (*Liu et al., 2023*). YT521-B homology domain family member 2 (YTHDF2), belonging to the YTH domain protein family, has been validated as m$^6$A 'reader' and regulates the stability of messenger RNAs (mRNAs) (*Chen et al., 2022*). YTHDF2 promotes the progression of lung adenocarcinoma by recognizing m$^6$A modification and influencing mRNA fate (*Li et al., 2021*). Furthermore, YTHDF2 orchestrates the reprogramming of tumor-associated macrophages in the tumor microenvironment (TME), and YTHDF2 is an effective target to enhance cancer immunotherapy (*Ma et al., 2023*). Functional analysis identifies that PDE1A is a promoter of NSCLC metastasis through its interaction with YTHDF2.

## Results

### PDE1A overexpression predicts a poor prognosis in lung cancer patients

First, immunohistochemistry analysis revealed that PDE1A expression was significantly higher in lung cancer tissues compared to normal lung tissues (*Figure 1A*, *Figure 1—figure supplement 1*; *Uhlén et al., 2015*; *Uhlen et al., 2017*). As shown in *Figure 1B*, overexpression of PDE1A was also observed in three NSCLC cell lines compared with normal human lung fibroblasts (HELF cells). Additionally, the overexpression of PDE1A was also observed in lung cancer from high-risk patients compared with low-risk patients (p<0.0001, *Figure 1C*). Lung cancer patients in the high-risk group had shorter survival times than those in the low-risk group (*Figure 1D*; *Aguirre-Gamboa et al., 2013*; *Chitale et al., 2009*). Furthermore, lung cancer patients with high levels of PDE1A in their tumors had shorter overall survival than those with low PDE1A expression, indicating that PDE1A overexpression was correlated with a poor prognosis in lung cancer patients (*Figure 1E*; *Rousseaux et al., 2013*; *Goswami and Nakshatri, 2014*). Thus, PDE1A might be a novel prognostic predictor in lung cancer treatment and contribute to lung cancer progression.

### PDE1A promotes the metastasis and EMT of NSCLC cells both in vitro and in vivo

To investigate the biological function of PDE1A in lung cancer development, gene set enrichment analysis (GSEA) and overrepresentation enrichment analysis (ORA) were performed to analyze the

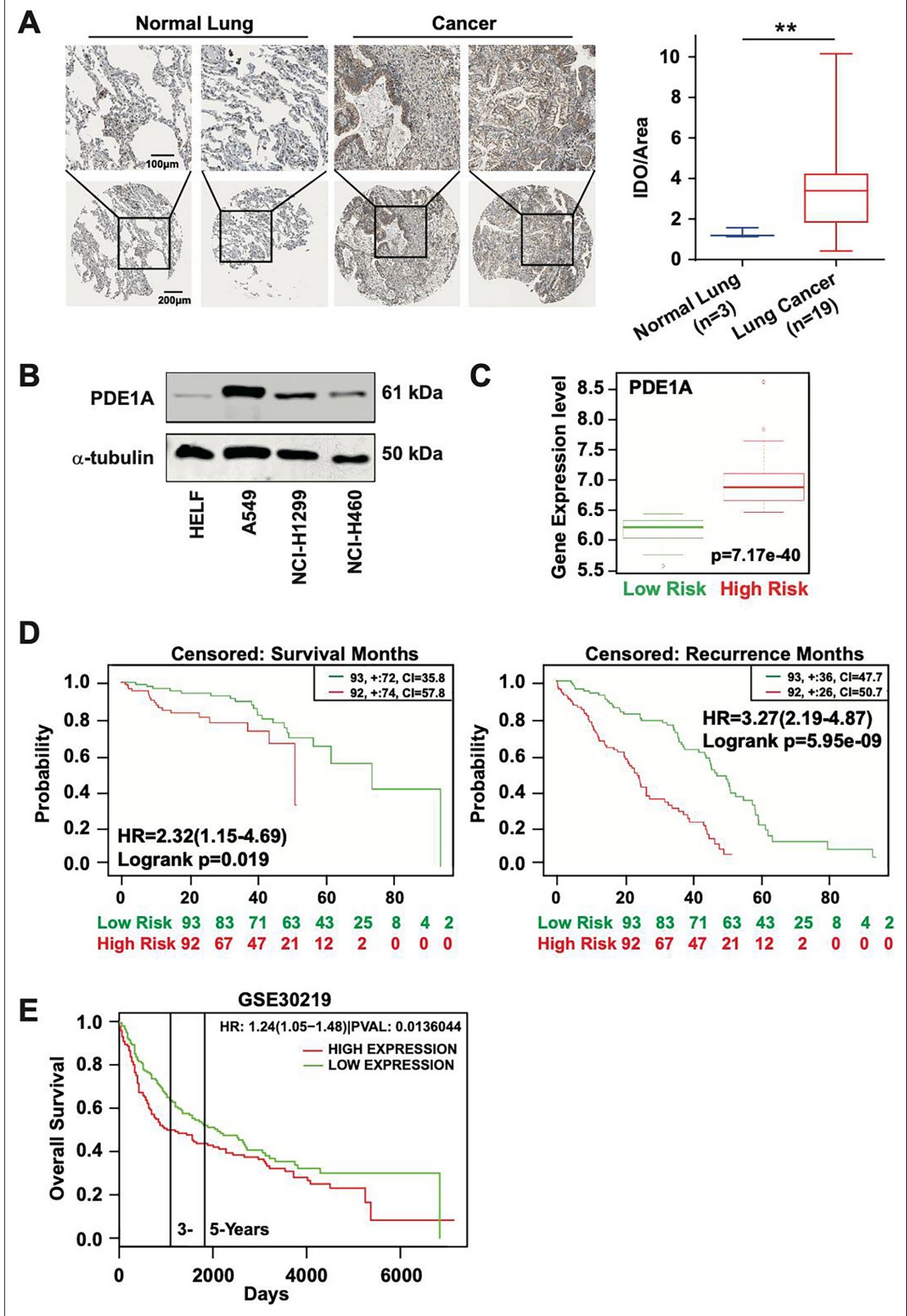

**Figure 1.** High expression of phosphodiesterase 1A (PDE1A) predicts a poor prognosis of lung cancer patients. (**A**) The expression of PDE1A was detected in non-small cell lung cancer (NSCLC) and normal lung tissue. It was obtained from The Human Protein Atlas (https://www.proteinatlas.org/). IOD/area means integral optical density/area. t-Test was used to compare the difference between lung cancer and normal lung groups, **$P < 0.01$. (**B**) The expression of PDE1A in human lung fibroblasts (HELF), A549, NCI-H1299, and NCI-H460 cells was detected by western blot. (**C and D**) Box plot

*Figure 1 continued on next page*

*Figure 1 continued*

analysis of the PDE1A messenger RNA (mRNA) levels in clinical lung cancer tissue samples. It was collected, and statistical analyses were performed from SurvExpress (http://bioinformatica.mty.itesm.mx:8080/Biomatec/SurvivaX.jsp). Gene: PDE1A; Access database numbers: Chitale lung (n=185); Censored: recurrence months or survival months. (**E**) It was collected from PROGgene V2 Prognostic Database (http://www.progtools.net/gene/index.php). Survival analysis is done using backend R script which employs R library 'survival' to perform Cox proportional hazards analysis (function 'coxph') and to plot prognostic plots (function 'survfit'). Single-user input genes: PDE1A; Cancer type: LUNG; Survival measure: death; Bifurcate gene expression at: median; GSE30219-Off-context gene expression in lung cancer identifies a group of metastatic-prone tumors.

The online version of this article includes the following source data and figure supplement(s) for figure 1:

**Source data 1.** Raw images for western blot shown in *Figure 1B* (labelled).

**Source data 2.** Raw images for western blot shown in *Figure 1B*.

**Figure supplement 1.** The expression of phosphodiesterase 1A (PDE1A) in normal lung (**A**) and lung cancer (**B**).

**Figure supplement 2.** The biological processes related to phosphodiesterase 1A (PDE1A) in non-small cell lung cancer (NSCLC) are predicted using LinkedOmics.

**Figure supplement 3.** The expression of phosphodiesterase 1A (PDE1A) does not affect the proliferation of non-small cell lung cancer (NSCLC) cells.

biological process of PDE1A in NSCLC using LinkedOmics (*Vasaikar et al., 2018*). As shown in *Figure 1—figure supplement 2A*, PDE1A might be involved in the adhesion, migration, and motility of NSCLC cells, which are critical parameters in the metastatic dissemination of cancer cells. Tumor angiogenesis, the recruitment of new blood vessels, enables a pre-existing tumor to grow and metastasize (*Onn and Herbst, 2003*). PDE1A might also participate in mesenchyme development, angiogenesis, vasculature development, cellular response to VEGF stimulus, blood vessel morphogenesis, and development (*Figure 1—figure supplement 2*). Thus, PDE1A is proposed to enhance the metastatic potential of NSCLC cells.

First, PDE1A silencing did not cause a significant decrease in the proliferation of NSCLC cells relative to that in the control siRNA group (*Figure 1—figure supplement 3A*). Moreover, PDE1A overexpression had no significant effect on the proliferation of NSCLC cells (*Figure 1—figure supplement 3B*). As bioinformatics analysis demonstrated that PDE1A might promote the metastatic potential of NSCLC cells, wound healing and Transwell assays were used to investigate the migration and invasion capacity of PDE1 family members. Knockdown of PDE1 family members suppressed the migratory ability of NCI-H1299 cells, and siPDE1A exerted a stronger suppression effect on the migration of NCI-H1299 cells than siPDE1B and siPDE1C transfection (*Figure 2—figure supplement 1A*). Meanwhile, siPDE1A resulted in more profound suppression of epithelial-mesenchymal transition (EMT) progression in NCI-H1299 cells than siPDE1B and siPDE1C transfection (*Figure 2—figure supplement 1B*). Thus, PDE1 family members, particularly PDE1A, might be involved in the metastatic behavior of NSCLC cells.

As shown in *Figure 2A and B*, suppression of PDE1A markedly reduced the migratory and invasive capacity of NSCLC cell lines. The wound healing assay also showed that NSCLC cells with PDE1A knockdown had a slower wound closure rate than control siRNA-transfected cells (*Figure 2C and D*). Furthermore, PDE1A knockdown increased E-cadherin expression and reduced N-cadherin expression, indicating that PDE1A silencing suppressed EMT progression in NSCLC cells (*Figure 2E*). Meanwhile, the PDE1 inhibitor vinpocetine significantly suppressed the migration and EMT of NSCLC cells (*Figure 2F–H*). To determine the effects of PDE1A on NSCLC cell migration and invasion in vivo, nude mouse models were established using PDE1A-shRNA- and control-shRNA-treated NCI-H1299 cells. As shown in *Figure 2I*, *Figure 2—figure supplement 1C and D*, the number of pulmonary metastatic nodules was decreased in the PDE1A-shRNA group compared with the control-shRNA group in nude mice.

In contrast, PDE1A overexpression significantly enhanced the migratory and invasive capacities of NSCLC cells (*Figure 3A and B*). In addition, NSCLC cells with high PDE1A expression had a higher wound closure rate than those transfected with empty vector (*Figure 3C*). Meanwhile, PDE1A overexpression decreased E-cadherin expression and elevated N-cadherin expression, indicating that PDE1A promoted EMT progression of NSCLC cells (*Figure 3D*). NSCLC cell lines with varying invasive potential were generated via repeated Transwell selection to compare PDE1A expression between highly and poorly invasive cells (*Figure 3E*, *Figure 3—figure supplement 1A and B*). The protein and mRNA levels of PDE1A were higher in highly invasive NSCLC cells than in NSCLC cells

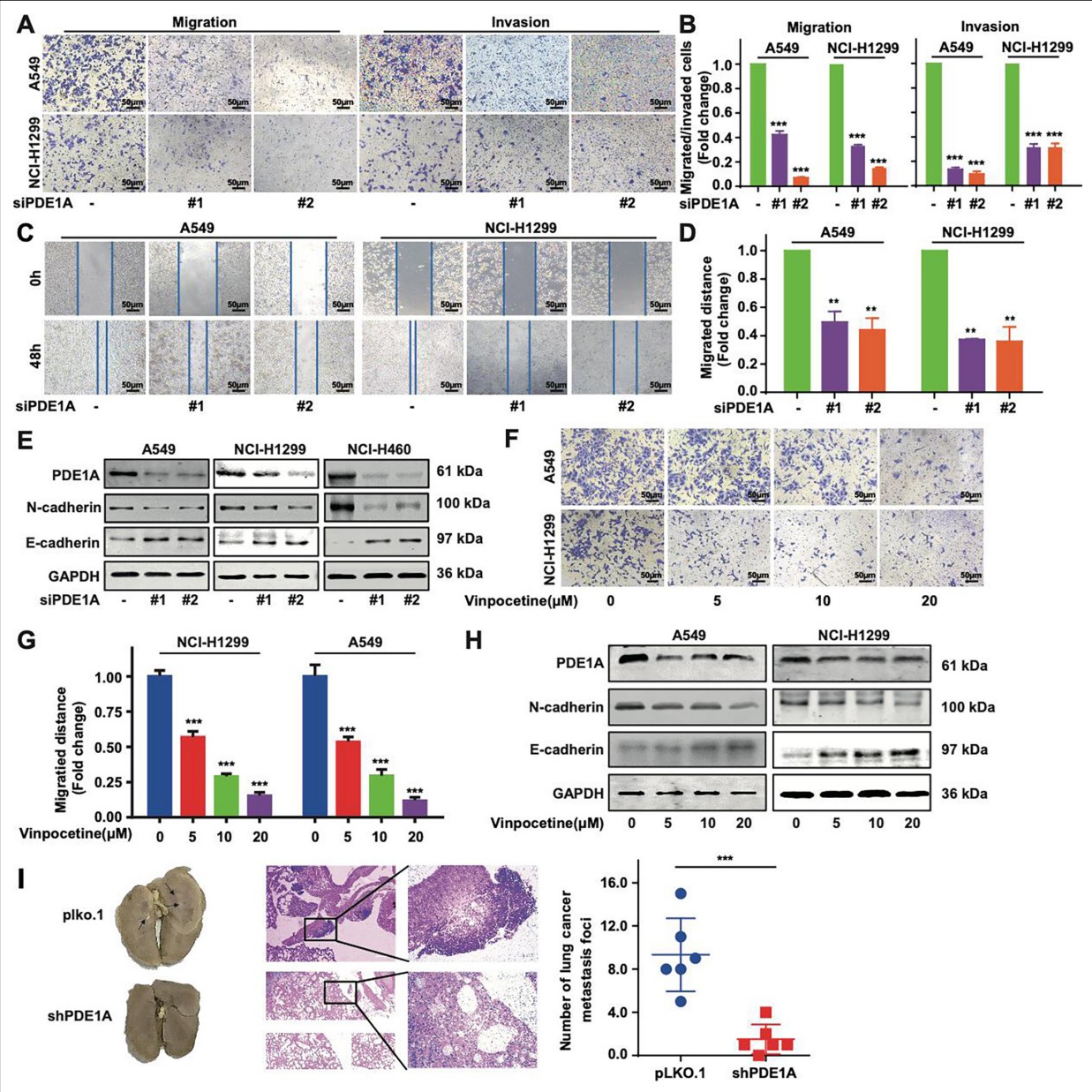

**Figure 2.** Phosphodiesterase 1A (PDE1A) knockdown suppresses the metastasis of non-small cell lung cancer (NSCLC) cells. (**A–B**) NSCLC cells were transfected with control siRNA and PDE1A siRNA for 24 hr. Cells were transferred to Transwell chambers without or with a Matrigel coating on the insert membrane, and the cell migrative and invasive abilities were determined, respectively (n=3). (**C–D**) NSCLC cells were transfected with control siRNA and PDE1A siRNA for 24 hr, and the wound healing assay was established in NSCLC cells (n=3). (**E**) NSCLC cells were transfected with control siRNA and PDE1A siRNA for 48 hr, and the expression of indicated proteins was detected. (**F–G**) NSCLC cells were treated with DMSO or vinpocetine (5, 10, 20 μM) for 24 hr, and the migrative ability of treated NSCLC cells was determined using the Transwell assay for 24 hr (n=3). (**H**) NSCLC cells were treated with DMSO or vinpocetine (5, 10, 20 μM) for 24 hr, and the expression of indicated proteins was determined. (**I**) The pulmonary metastatic nodules were stained using H&E staining and counted in nude mice harboring NCI-H1299 cells transfected with PDE1A shRNA and control shRNA (n=6). $^{**}P < 0.01$, $^{***}P < 0.001$.

The online version of this article includes the following source data and figure supplement(s) for figure 2:

**Source data 1.** Raw images for western blots shown in *Figure 2* (labelled).

**Source data 2.** Raw images for western blots shown in *Figure 2*.

**Figure supplement 1.** Phosphodiesterase 1A (PDE1A) silence suppresses the metastasis of non-small cell lung cancer (NSCLC) cells.

**Figure supplement 1—source data 1.** Raw images for western blot shown in *Figure 2—figure supplement 1B* (labelled).

**Figure supplement 1—source data 2.** Raw images for western blot shown in *Figure 2—figure supplement 1B*.

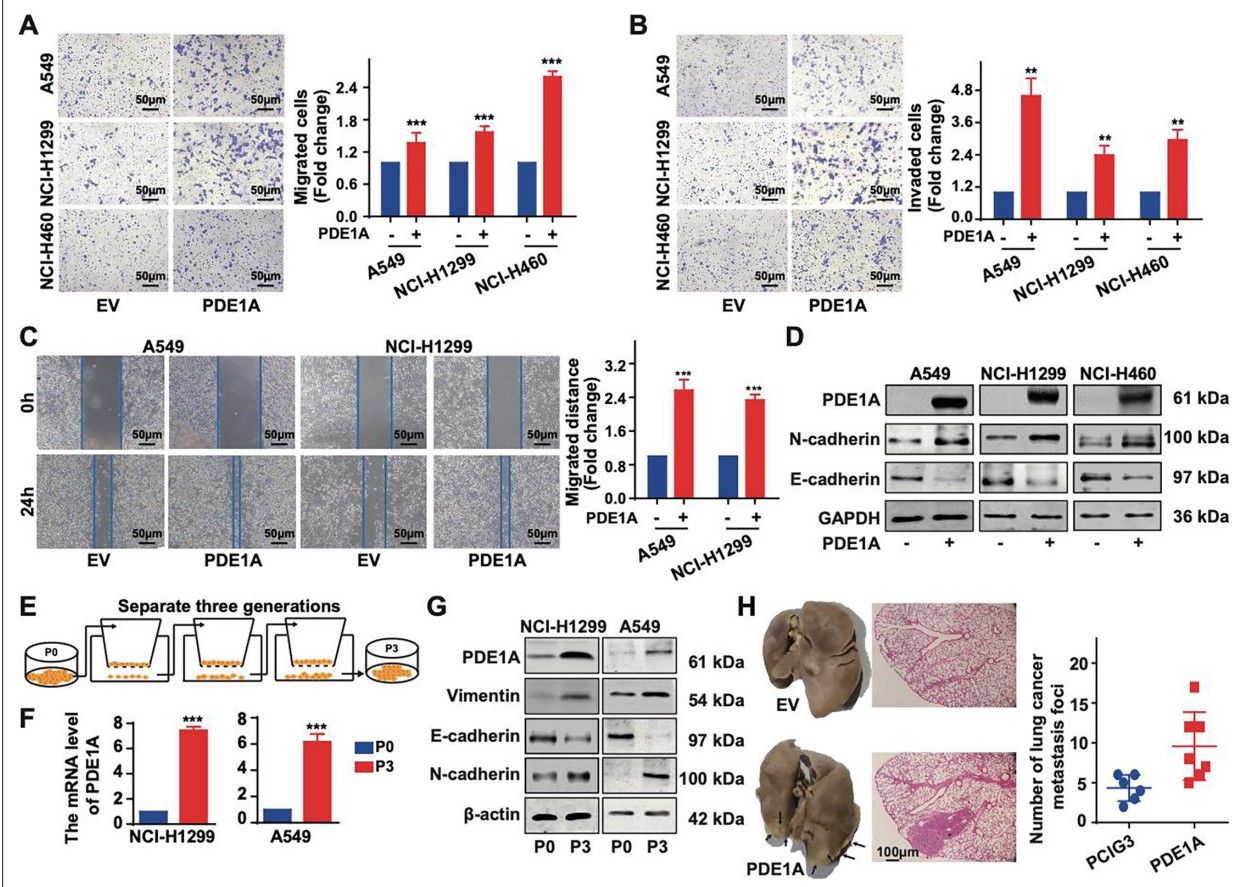

**Figure 3.** Phosphodiesterase 1A (PDE1A) promotes the metastasis and epithelial-mesenchymal transition (EMT) of non-small cell lung cancer (NSCLC) cells. (**A–B**) NSCLC cells were transfected with PDE1A plasmid and empty vector for 24 hr. Cells were transferred to Transwell chambers without (**A**) or with (**B**) a Matrigel coating on the insert membrane, and the cell migrative and invasive abilities were determined, respectively, (n=3). (**C**) NSCLC cells were transfected with PDE1A plasmid and empty vector for 24 hr, and the wound healing assay was established in NSCLC cells, (n=3). (**D**) NSCLC cells were transfected with PDE1A plasmid and empty vector for 48 hr, and the expression of indicated proteins was detected. (**E**) The highly invasive NSCLC cells were separated using the Transwell chamber assay, and P3 cells were obtained from P0 cells after three generations. (**F–G**) The messenger RNA (mRNA) (**F**) and protein (**G**) levels of indicated genes were determined in P3 and P0 NSCLC cells. (**H**) The pulmonary metastatic nodules were stained using H&E and Bouin's solution and counted in nude mice harboring NCI-H1299 cells transfected with PDE1A plasmid and empty vector, (n=6). $^{**}P < 0.01$, $^{***}P < 0.001$.

The online version of this article includes the following source data and figure supplement(s) for figure 3:

**Source data 1.** Raw images for western blots shown in *Figure 3* (labelled).

**Source data 2.** Raw images for western blots shown in *Figure 3*.

**Figure supplement 1.** Phosphodiesterase 1A (PDE1A) promotes the metastasis of non-small cell lung cancer (NSCLC) cells.

**Figure supplement 1—source data 1.** Raw images for western blots shown in *Figure 3—figure supplement 1C* (labelled).

**Figure supplement 1—source data 2.** Raw images for western blots shown in *Figure 3—figure supplement 1C*.

with low invasive potential (*Figure 3F and G*). In an in vivo nude mouse experiment, NSCLC cells overexpressing PDE1A produced more pulmonary metastatic nodules than the parental NSCLC cells (*Figure 3H*, *Figure 3—figure supplement 1C and D*).

## NSCLC cells overexpressing PDE1A promote angiogenesis in the TME

GSEA demonstrated that PDE1A expression was positively correlated with angiogenesis in lung cancer (*Figure 4A*). To mimic the TME, a coculture system of NSCLC cells and vascular endothelial cells was established (*Figure 4B*). NSCLC cells overexpressing PDE1A promoted the migration of human umbilical vein endothelial cells (HUVECs), and NSCLC cells with low levels of PDE1A suppressed the migration of HUVECs (*Figure 4C and D*). Next, NSCLC cells were treated with GW4869 to reduce

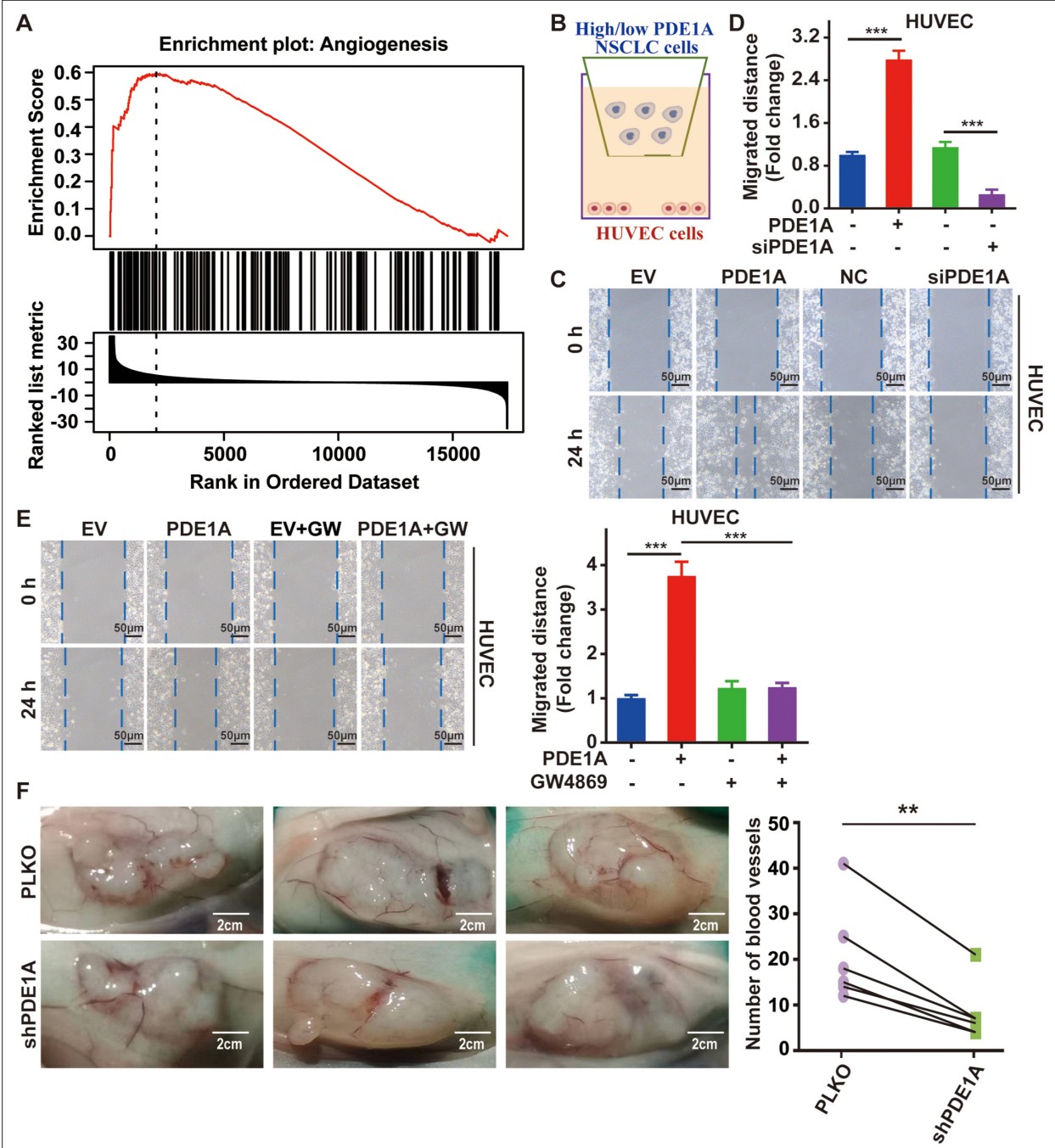

**Figure 4.** Non-small cell lung cancer (NSCLC) cells overexpressing phosphodiesterase 1A (PDE1A) promote angiogenesis in the tumor microenvironment (TME). (**A**) It was collected from LinkedOmics. Statistical tests in LinkFinder include Pearson's correlation coefficient, Spearman's rank correlation, Student's t-test, Wilcoxon test, analysis of variance, Kruskal-Wallis analysis, Fisher's exact test, Chi-squared test, Jonckheere's trend test, and Cox's regression analysis. Multiple-test correction is performed using the Benjamini and Hochberg method to generate the false discovery rate. (**B–D**) NSCLC cells were transfected with empty vector/PDE1A overexpressing plasmid or control siRNA/siPDE1A for 48 hr, and then NSCLC cells were placed on the upper panel of Transwell with 0.4 μm insert, human umbilical vein endothelial cells (HUVECs) were placed on the lower panel of Transwell, and wound healing assay was performed to determine the migrative abilities of HUVECs, (n=3). (**E**) NSCLC cells with PDE1A overexpression were treated with 10 μM GW4869, and a wound healing assay was performed to determine the migrative abilities of HUVECs, (n=3). (**F**) NSCLC cells were transfected with empty vector or shPDE1A, then cells were transplanted into nude mice via subcutaneous injection, and the blood vessels were counted after 60 days, (n=6). **$P < 0.01$, ***$P < 0.001$.

exosome release, and GW4869 suppressed the enhancement of the migratory ability of HUVECs induced by NSCLC cells overexpressing PDE1A (*Figure 4E*). Meanwhile, compared with negative control, shPDE1A significantly suppressed tumor angiogenesis of NSCLC in vivo (*Figure 4F*). Thus, NSCLC cells overexpressing PDE1A promote angiogenesis in the TME.

## PDE1A promotes the metastasis of NSCLC cells via the STAT3 signaling pathway

Then, the dependence of PDE1A-enhanced metastasis on cAMP metabolic activity was investigated. As shown in *Figure 5—figure supplement 1*, the cAMP inhibitor SQ22536 failed to rescue the migrative ability suppressed by siPDE1A in NSCLC cells, indicating that the basic molecular function might not be involved in the metastasis of PDE1A. To better explore the mechanism of PDE1A in NSCLC progression, bioinformatic analysis of PDE1A coexpressed genes was performed, which revealed that PDE1A might be involved in the JAK/STAT3, Hedgehog, and TGF-β pathways in NSCLC (*Figure 5A*). Meanwhile, GSEA enrichment analysis demonstrated that PDE1A might participate in IL-6 production (*Figure 5B*). Thus, IL-6/JAK/STAT3 signaling is involved in PDE1A-mediated promotion of metastasis in NSCLC. As shown in *Figure 5C*, PDE1A overexpression increased the phosphorylation level of STAT3 in NSCLC cells. In contrast, PDE1A knockdown or the PDE1 inhibitor vinpocetine suppressed the phosphorylation of STAT3 in NSCLC cells (*Figure 5D and E*). Moreover, STAT3 suppression by siRNA or SH-4–54 significantly reversed the enhancement of NSCLC cell migration induced by PDE1A overexpression (*Figure 5F and G*). In addition, the suppression of STAT3 inhibited PDE1A-induced EMT progression in NSCLC cells (*Figure 5H and I*). Thus, PDE1A promoted the metastasis of NSCLC cells via activating the STAT3 signaling pathway, but the direct interaction between PDE1A and STAT3 could not be observed in NSCLC cells (*Figure 5J*). Moreover, PDE1A was mainly overexpressed in the cytoplasm in NSCLC cells (*Figure 5K*). Subsequently, the mechanism by which PDE1A promotes the STAT3 signaling pathway in the cytoplasm was further explored.

## PDE1A physically interacts with YTHDF2 and promotes the metastasis of NSCLC cells

To investigate the mechanism by which PDE1A promotes NSCLC metastasis and activates the STAT3 pathway, the proteins interacting with PDE1A in NSCLC were determined using immunoprecipitation followed by mass spectrometry analysis (*Supplementary file 3*). To identify key proteins involved in PDE1A-mediated STAT3 activation, a Venn analysis revealed nine overlapping genes among STAT3-coexpressed genes in NSCLC samples, PDE1A-interacting proteins, and genes overexpressed in NSCLC compared to normal tissues (*Figure 6A*). Meanwhile, GO enrichment analysis of PDE1A-interacting genes was used to predict the molecular function of PDE1A, and PDE1A might participate in m⁶A-containing RNA binding in NSCLC progression (*Figure 6B*). Based on this, it was hypothesized that PDE1A may interact with YTHDF2 and be involved in the binding of m⁶A-modified RNA during NSCLC progression. The physical binding between PDE1A and YTHDF2 was confirmed by silver staining and immunoprecipitation (*Figure 6C and D*). Furthermore, YTHDF2 knockdown reversed the enhancement of NSCLC migration induced by PDE1A overexpression, indicating that PDE1A might interact with YTHDF2 and promote the metastasis of NSCLC (*Figure 6E*, *Figure 6—figure supplement 1A and B*). The mRNA and protein levels of YTHDF2 were upregulated in NSCLC compared with normal lung tissues (*Figure 6—figure supplement 1C and D*; *Chandrashekar et al., 2017*). In addition, YTHDF2 overexpression predicted poor outcomes in lung cancer patients (*Figure 6—figure supplement 1E*; *Rousseaux et al., 2013*; *Goswami and Nakshatri, 2014*). Meanwhile, the activation of STAT3 by PDE1A could be reversed by YTHDF2 knockdown in NSCLC cells (*Figure 6F*). Furthermore, PDE1A was positively correlated with YTHDF2 expression with a Pearson's correlation coefficient above 0.3 in NSCLC tissues (*Figure 6G*; *Chandrashekar et al., 2017*). Thus, PDE1A might regulate the STAT3 signaling pathway via interacting with YTHDF2.

## PDE1A interacts with YTHDF2 to regulate the SOCS2/STAT3 signaling pathway

To investigate how the PDE1A/YTHDF2 complex regulates STAT3 signaling, Venn analysis identified 33 genes overlapping among YTHDF2-bound RNAs and genes coexpressed with YTHDF2 and STAT3 in lung cancer (*Shi et al., 2017*; *Bartha and Győrffy, 2021*; *Cerami et al., 2012*; *Gao et al., 2013*;

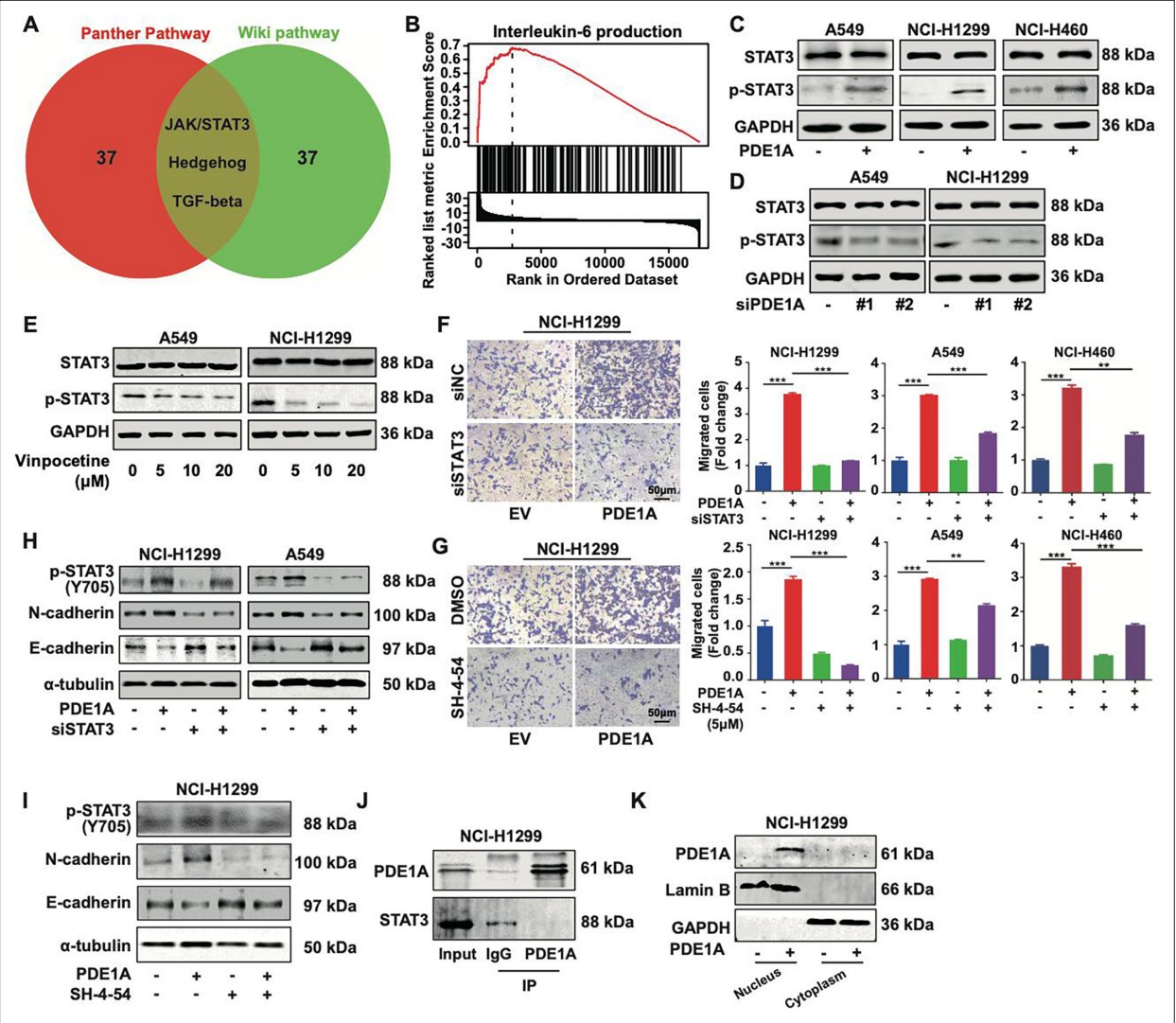

**Figure 5.** Phosphodiesterase 1A (PDE1A) promotes the metastasis of non-small cell lung cancer (NSCLC) cells via the STAT3 signaling pathway. (**A**) A Venn diagram was generated using LinkedOmics, and overrepresentation enrichment analysis (ORA) was performed to analyze the molecular pathway regulated by PDE1A in NSCLC. Sample cohort: TCGA_NSCLC; Institute: UNC; Data type: RNAseq; Platform: HiSeq RNA; Attribute: PDE1A; Statistical methods: Spearman's correlation test; Patients: 515; Tools: ORA; Gene Ontology analysis: WikiPathways and PANTHER Pathway; Select rank criteria: FDR; Select sign: Positively correlated; Significance level: 0.05; TOP40 was selected to generate the Venn diagram. (**B**) Gene set enrichment analysis (GSEA) was performed to analyze the biological process of PDE1A in NSCLC. (**C**) NSCLC cells were transfected with PDE1A plasmid and empty vector for 48 hr, and the expression of indicated proteins was detected. (**D**) NSCLC cells were transfected with control siRNA and PDE1A siRNA for 48 hr, and the expression of indicated proteins was detected. (**E**) NSCLC cells were treated with DMSO or vinpocetine (5, 10, 20 µM) for 24 hr, and the expression of indicated proteins was determined. (**F**) NSCLC cells overexpressing PDE1A were transfected with control siRNA and STAT3 siRNA for 48 hr, and the migrative abilities of NSCLC cells were determined by Transwell assay, (n=3). (**G**) NSCLC cells overexpressing PDE1A were treated with STAT3 inhibitor SH-4–54 (5 µM) for 24 hr. The migrative abilities of NSCLC cells were determined by Transwell assay, (n=3). (**H**) NSCLC cells overexpressing PDE1A were transfected with control siRNA and STAT3 siRNA for 48 hr, and the expression of indicated protein was detected by western blot. (**I**) NSCLC cells overexpressing PDE1A were treated with STAT3 inhibitor SH-4–54 (5 µM) for 24 hr, and the expression of indicated protein was detected by western blot. (**J**) The interaction between PDE1A and STAT3 was determined by immunoprecipitation. (**K**) NCI-H1299 cells were transfected with empty vector and PDE1A overexpressing plasmid for 48 hr, and the contribution of PDE1A in the cytoplasm and nucleus was determined. $^{**}P < 0.01$, $^{***}P < 0.001$.

The online version of this article includes the following source data and figure supplement(s) for figure 5:

**Source data 1.** Raw images for western blots shown in *Figure 5* (labelled).

**Source data 2.** Raw images for western blots shown in *Figure 5*.

**Figure supplement 1.** Phosphodiesterase 1A (PDE1A) promotes the metastatic potential in a cyclic adenosine monophosphate (cAMP)-independent manner.

*Figure 5 continued on next page*

*Figure 5 continued*

**Figure supplement 1—source data 1.** Raw images for western blots related to *Figure 5—figure supplement 1* (labelled).

**Figure supplement 1—source data 2.** Raw images for western blots related to *Figure 5—figure supplement 1*.

*Figure 7A* and *Supplementary file 4*). Then, the interactions between the YTHDF2 protein and the mRNAs of 33 overlapping genes were predicted by the RNA-Protein Interaction Prediction online tool. There were three predicted targets of YTHDF2 with high scores and highly correlated with STAT3 signaling as previously reported, including NRF2, SOCS2, and MET (*Supplementary file 5*; *Muppirala et al., 2011*). SOCS family members are cytokine-inducible negative regulators of the JAK/STAT pathway, and SOCS2 suppresses the binding of JAK2 and STAT3, the activity of JAK, and STAT3 activation (*Sen et al., 2012*). It was hypothesized that PDE1A might interact with YTHDF2, affect the stability of *SOCS2* mRNA, and thereby regulate the STAT3 signaling pathway in NSCLC cells. As shown in *Figure 7B*, the interaction between YTHDF2 protein and SOCS2 mRNA was confirmed by RIP, and the binding between PDE1A protein and SOCS2 mRNA was also demonstrated using RIP. Meanwhile, siPDE1A significantly enhanced the stability of SOCS2 mRNA in NSCLC cells (*Figure 7C*). Furthermore, YTHDF2 or PDE1A negatively regulated the expression of SOCS2 mRNA in NSCLC cells, and YTHDF2 overexpression successfully reversed the siPDE1A-induced SOCS2 mRNA accumulation. In contrast, siYTHDF2 enhanced siPDE1A-induced SOCS2 mRNA accumulation (*Figure 7D*). Thus, YTHDF2 might negatively regulate the expression of SOCS2 mRNA via cooperating with PDE1A.

## Discussion

NSCLC is becoming a leading cause of death globally due to its fast progression and metastatic potential, and effective therapeutic targets are urgently needed to block NSCLC metastasis (*Zhu et al., 2020*). PDEs are regarded as therapeutic targets for multiple diseases, but the feasibility of targeting PDEs to treat NSCLC metastasis may need further investigation (*Chen and Yan, 2021*). It is the first evidence that PDE1 promotes metastasis and EMT progression in NSCLC cells. Furthermore, the expression of PDE1A was closely correlated with the disease progression of NSCLC. Thus, PDE1A might be an efficacious therapeutic target for patients with metastatic NSCLC.

The stimulation of the PDEs requires physiological concentrations of $Ca^{2+}$ and calmodulin, and $Ca^{2+}$/calmodulin-dependent cyclic nucleotide PDE1 is involved in the communication between the cyclic nucleotide and $Ca^{2+}$ second messenger systems (*Kakkar et al., 1999*). The increasing of $Ca^{2+}$ in cancer cells stimulates exosome biogenesis and release under both physiological and pathological conditions (*Han et al., 2022*; *Messenger et al., 2018*). It was hypothesized that PDE1A might be involved in exosome biogenesis and release in NSCLC cells, playing a crucial role in intercellular communication within the TME. GSEA showed that PDE1A might be involved in angiogenesis, vasculature development, and blood vessel development. Indeed, NSCLC cells overexpressing PDE1A promoted angiogenesis in the TME, and PDE1A knockdown significantly suppressed angiogenesis of NSCLC in vivo. Furthermore, an exosome release inhibitor successfully reversed the angiogenesis promoted by NSCLC cells overexpressing PDE1A. Thus, PDEs might play an important role in angiogenesis and the TME via regulating exosome biogenesis and release in cancer cells. Analysis of PDE1A co-expressed genes in NSCLC revealed a significant enrichment of the IL-6/JAK/STAT3 signaling pathway, suggesting its involvement as a downstream pathway of PDE1A. Furthermore, PDE1A promoted the metastasis of NSCLC cells via the STAT3 signaling pathway. Targeting the IL-6/JAK/STAT3 signaling pathway is considered a promising therapeutic strategy for the management of NSCLC (*Mohrherr et al., 2020*). However, the direct interaction between PDE1A and STAT3 could not be observed in NSCLC cells. Subsequently, the mechanism by which PDE1A promotes the STAT3 signaling pathway was investigated. It demonstrated that PDE1A interacts with YTHDF2 and contributes to NSCLC progression, with the interaction between YTHDF2 and PDE1A being verified for the first time in NSCLC cells. Meanwhile, YTHDF2 might act as an m6A RNA 'reader' by interacting with PDE1A, but the mechanism might need further investigation. YTHDF2 destabilizes mRNAs via degrading target transcripts, but it also stabilizes important oncogenic drivers, such as *MYC* and *VEGFA* transcripts, in an $m^6A$-dependent manner (*Dixit et al., 2021*). It was demonstrated that YTHDF2 destabilized SOCS2 mRNA via interacting with PDE1A, but the mechanism by which YTHDF2 sorts mRNA might need further investigation. In addition, it is worth testing if PDE1A

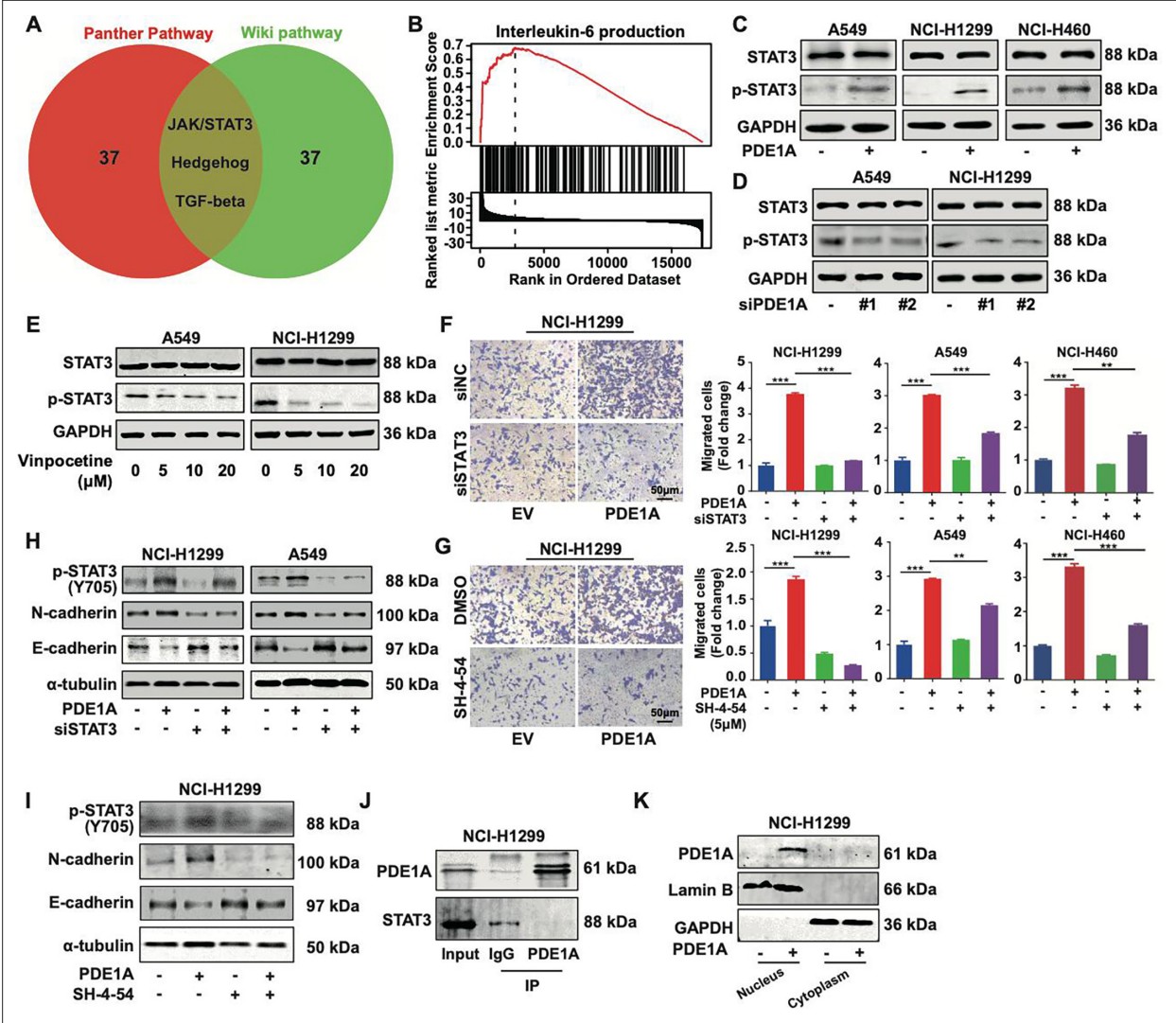

**Figure 6.** PDE1A physically interacts with YT521-B homology domain family member 2 (YTHDF2) and promotes the metastasis of non-small cell lung cancer (NSCLC) cells. (**A**) Venn diagram showing the overlap among PDE1A-interacting proteins (it was collected from mass spectrometry analysis in NSCLC cells), STAT3-coexpressed genes (collected from gene correlation using UALCAN), upregulated proteins in NSCLC compared with normal tissues (analyzed by UALCAN based on CPTAC database), and upregulated genes in NSCLC compared with normal tissues (analyzed by UALCAN based on TCGA database). Pearson's correlation analysis of UALCAN was used to evaluate gene correlation analyses, and Welch's t-test was estimated to detect the significance of differences in expression levels between two groups. (**B**) GO enrichment analysis of PDE1A-interacting genes. (**C**) Immunoprecipitation followed by silver staining was performed to identify protein and protein interaction using A549 cell lysate with the anti-PDE1A antibody. (**D**) Immunoprecipitation was used to confirm protein and protein interaction in NCI-H1299 cells. (**E**) NSCLC cells overexpressing PDE1A were transfected with control siRNA and YTHDF2 siRNA for 48 hr, and the migrative abilities of NSCLC cells were determined by Transwell assay, (n=3). (**F**) NSCLC cells overexpressing PDE1A were transfected with control siRNA and YTHDF2 siRNA for 48 hr, and the expression of indicated protein was detected by western blot. ***P < 0.001.

The online version of this article includes the following source data and figure supplement(s) for figure 6:

**Source data 1.** Raw images for western blots shown in *Figure 6* (labelled).

**Source data 2.** Raw images for western blots shown in *Figure 6*.

**Figure supplement 1.** YT521-B homology domain family member 2 (YTHDF2) predicts poor outcomes for lung cancer patients.

**Figure supplement 1—source data 1.** Raw images for western blots shown in *Figure 6—figure supplement 1A* (labelled).

**Figure supplement 1—source data 2.** Raw images for western blots shown in *Figure 6—figure supplement 1A*.

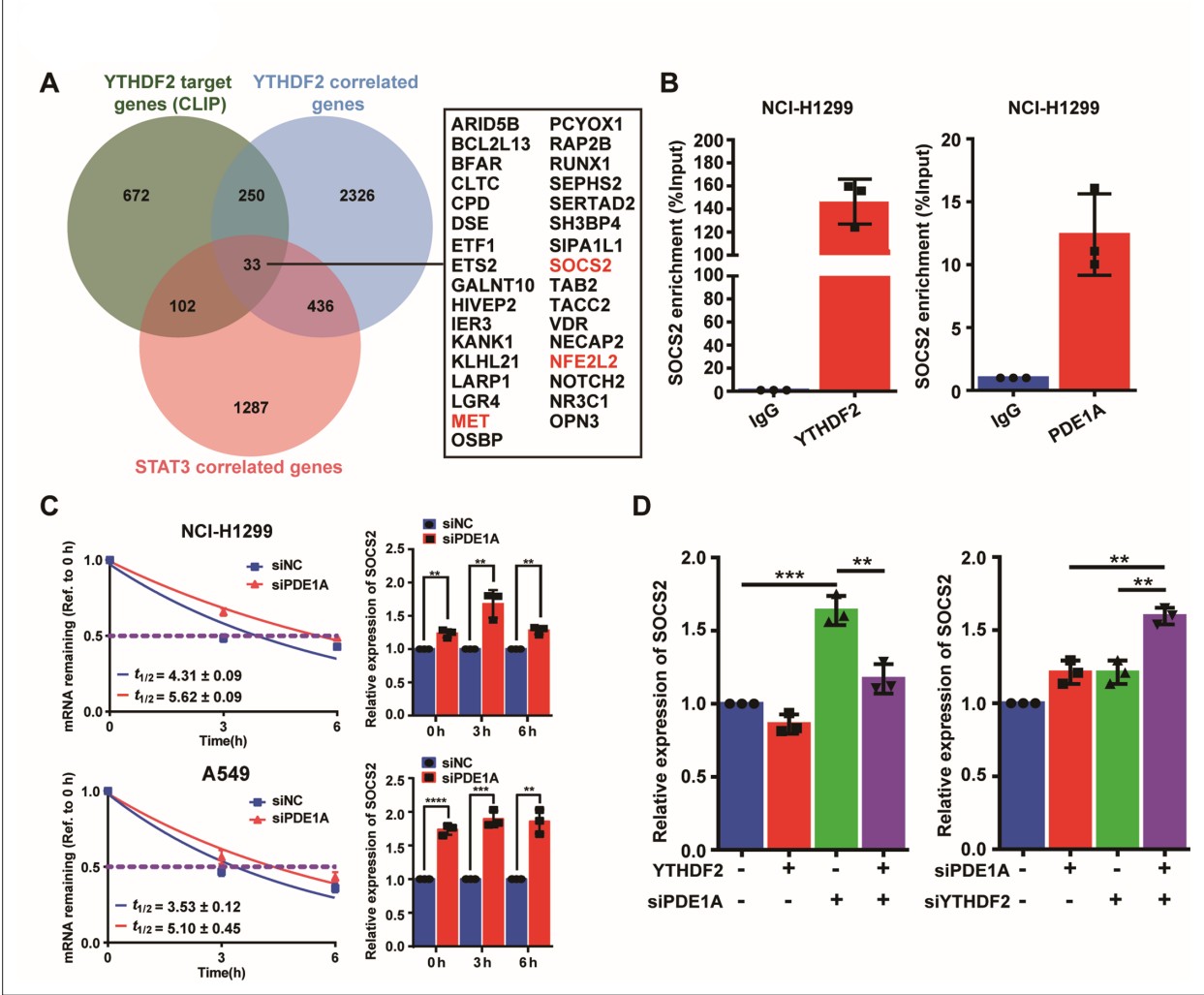

**Figure 7.** Phosphodiesterase 1A (PDE1A) interacts with YT521-B homology domain family member 2 (YTHDF2) to regulate SOCS2/STAT3 signaling pathway. (**A**) YTHDF2-RNA complexes were identified by LC-MS/MS and collected from reference; YTHDF2 correlated genes were collected from TNMplot (https://tnmplot.com/analysis/), Gene: YTHDF2, Gene vs. all genes correlation: Genechip data, Tissue: Lung; STAT3 correlated genes were collected from cBioPortal (http://www.cbioportal.org/), Lung cancer (SMC, cancer research 2016); n=22; Gene: STAT3. The interaction between YTHDF2 protein and the messenger RNA (mRNA) of 33 overlapping genes was predicted by RNA-Protein Interaction Prediction (http://pridb.gdcb.iastate.edu/RPISeq/index.html), and the values of RF classifier and SVM classifier above 0.5 were considered positive. Comparison of the normal and the tumorous samples was performed by the Mann-Whitney U test, and normal, tumorous, and metastatic tissue gene comparison can be analyzed using the Kruskal-Wallis test. (**B**) The interactions between protein and mRNAs were verified by RNA binding protein immunoprecipitation (RIP) experiments. (**C**) Non-small cell lung cancer (NSCLC) cells were transfected with control siRNA and siPDE1A for 48 hr, and the stability of mRNA was determined by quantitative real-time PCR (qRT-PCR), (n=3). (**D**) NSCLC cells were transfected with control siRNA and siPDE1A for 48 hr, and the expression of SOCS2 mRNA was determined by qRT-PCR, (n=3). $^{**}P < 0.01$, $^{***}P < 0.001$.

inhibition affects metastasis in lung cancer models and sensitizes cisplatin in resistant NSCLC cells in vitro and in vivo. The role of YTHDF2 in PDE1A-driven tumor metastasis should be elucidated in future studies.

Collectively, PDE1A promotes metastasis in NSCLC cells, and PDE1A overexpression is correlated with angiogenesis and poor outcomes of NSCLC patients. In addition, PDE1A interacts with YTHDF2 and regulates the JAK/STAT3 signaling pathway via degrading SOCS2 mRNA. Therefore, it reveals the effect and mechanism of PDE1A in promoting NSCLC metastasis. It not only uncovers a novel PDE1A/YTHDF2/STAT3 signaling pathway in NSCLC progression but also provides novel therapeutic strategies to treat NSCLC patients with metastasis.

# Materials and methods

## Materials

Vinpocetine (V107535) was purchased from Aladdin (Shanghai, China). SH-4–54 (S7337), SQ22536 (S8283), and GW4869 (S7609) were obtained from Selleck Chemicals (Houston, TX, USA). Antibodies against β-actin (SC-1616) and GAPDH (SC-25778) were obtained from Santa Cruz Biotechnology (Dallas, TX, USA). Antibodies against N-cadherin (14215), STAT3 (9139), p-STAT3 (Y-705) (9145), Vimentin (5741), and JAK2 (3230S) were obtained from Cell Signaling Technology (Danvers, MA, USA). An antibody against α-tubulin (AT7819) was purchased from Beyotime Biotechnology (Shanghai, China). Antibodies against PDE1A (12442-2-AP), YTHDF2 (247441-AP), lamin B (12595-1-AP), and E-cadherin (20874-1-AP) were obtained from Proteintech (Rosemont, IL, USA).

## Cell lines and cell culture

Human NSCLC cell lines (A549, NCI-H1299, and NCI-H460), HELF, and HUVECs were maintained in RPMI-1640 medium supplemented with 10% fetal bovine serum (FBS) and 100 U/ml penicillin/streptomycin. All the cell lines were purchased from the Shanghai Institute of Biochemistry and Cell Biology (Shanghai, China), cultured at 37°C with 5% $CO_2$ and confirmed to be mycoplasma-free.

## siRNA transfection

Scramble siRNA, siPDE1A, siPDE1B, siPDE1C, siSTAT3, and siYTHDF2 were synthesized by Gene-Pharma (Suzhou, China). Then, NSCLC cells were transfected with siRNA (40 nM) using PolyPlus-transfection reagent in accordance with the instructions. The sequences of siRNAs are summarized in *Supplementary file 1*.

## SRB assay

NSCLC cells were treated with the indicated compounds and subsequently fixed with ice-cold TCA and stained with 0.4% SRB (wt/vol) solution. Cell proliferation was determined by SRB assay according to the previously reported methods (*Hu et al., 2020*).

## Wound healing assay

NSCLC cells were seeded on a 24-well plate and cultured as a monolayer to 90% confluence. The monolayer was scratched with a 10 μl pipette tip, and then the cells were cultured with FBS-free culture medium for 24 hr. Images of the wounded cell monolayer were taken using a microscope (Olympus, Japan) at 0 and 24 hr. The wound closure rate was calculated as follows: $(G_0 - G_{24})/G_0 \times 100\%$, where $G_0$ and $G_{24}$ represent the gap areas at 0 and 24 hr, respectively.

## Migration and invasion assays using Transwell

Migration and invasion assays were performed using a 24-well Transwell chamber system (pore size: 8 μm, Corning, USA). A total of $5 \times 10^4$ cells were seeded in the upper chamber of an insert with 0.4 ml serum-free culture media in 24-well plates. Then, 0.6 ml culture medium with 20% FBS was added to the lower chamber. For invasion assays, the upper transwell chamber of the insert was coated with Matrigel (BD Biosciences, Bedford, MA, USA) before plating cells. 50 μl of Matrigel was dissolved in 450 μl of culture medium and added 100 μl solution into the upper transwell chamber. After incubation for 24 hr, migratory or invasive cells were stained with 0.5% crystal violet and analyzed under a light microscope.

## Protein extraction and western blotting analysis

Briefly, cells were washed three times with cold PBS and pelleted. The pellet was resuspended in lysis buffer (NP40 lysate), incubated on ice with frequent vortexing for 30 min, and the lysate was obtained by centrifugation at 10,000×*g* for 30 min. Proteins were fractionated by SDS-PAGE, transferred onto PVDF membranes, blocked in 5% nonfat milk in PBS/Tween-20, and then blotted with specific primary antibody overnight at 4°C, followed by incubation with secondary antibody for 1 hr at room temperature. Bands were detected by the Odyssey CLX Image Studio system (version 5.0.21, LiCor Odyssey, LI-COR Biosciences).

## Immunoprecipitation and LC-MS/MS

For co-immunoprecipitation, the cells were lysed with IP buffer (20 mM HEPES, 25% glycerine, 210 mM NaCl, 1.5 mM MgCl$_2$, 0.05 mM EDTA, 0.2% NP40, 1× cocktail, 1 mM PMSF, 2 mM DTT) and centrifuged at 10,000×$g$ for 30 min at 4°C. The cell lysates were treated with protein G magnetic beads at 4°C for 1 hr. Subsequent immunoprecipitation reactions were set up with equal quantities of the lysates. Primary antibody was added to the lysate, and the mixture was incubated overnight with slow shaking at 4°C, and then incubated with protein G magnetic beads at 4°C for 1 hr. Subsequently, the lysates were centrifuged at 3000×$g$ at 4°C for 5 min. The supernatant was aspirated, the protein G magnetic beads were washed three to four times with lysis buffer, and detection was performed using SDS-PAGE or LC-MS/MS (Micrometer Biotech Company, Hangzhou, China).

## Extraction of RNA and qRT-PCR

Total RNA was extracted from cells by utilizing TRIzol Reagent (Invitrogen, Thermo Fisher Scientific). RNA was reverse-transcribed into cDNA using an iScript cDNA Synthesis kit (Bio-Rad, Hercules, CA, USA). qRT-PCR was performed using the PerfectStart Green qPCR SuperMix kit (TransGen Biotech, Beijing, China). The levels of mRNA were analyzed by the Bio-Rad CFX96 real-time PCR system (Bio-Rad, Hercules, CA, USA). GAPDH was used as an endogenous control for mRNA qualification, and the $2^{-\Delta\Delta Ct}$ method was applied to calculate the relative expression. The primers used are listed in *Supplementary file 2*.

## RNA binding protein immunoprecipitation

RNA binding protein immunoprecipitation assays were performed using a Magna RIP Kit (17-701, Millipore, MA, USA) following the manufacturer's instructions. In brief, magnetic beads precoated with 5 μg normal antibodies against PDE1A/YTHDF2 or rabbit IgG (Millipore) were incubated with cell lysates at 4°C overnight. The beads containing immunoprecipitated RNA-protein complexes were treated with proteinase K to remove proteins. Then, RNAs of interest were purified with TRIzol and measured by qRT-PCR.

## Plasmid and shRNA transfection and infection

Two micrograms of overexpressing plasmid or shRNA of the indicated genes was transfected into cells using PolyPlus-transfection reagent. For shRNA used in lentivirus-mediated interference, complementary sense and antisense oligonucleotides encoding shRNAs targeting PDE1A were synthesized, annealed, and cloned into pLKO.TRC vector (Addgene, 10878). The PDE1A-FLAG overexpression plasmid was synthesized by GenScript (Nanjing, Jiangsu, China). YTHDF2-HA expression plasmids were synthesized via cloning YTHDF2 with an HA tag into the pcDNA3.1(-) vector.

## In vivo animal experiment

Female nude mice (BALB/c, 4–6 weeks of age) were obtained from Shanghai SLAC Laboratory Animal Co., Ltd. A total of 2×10$^6$ NCI-H1299 cells transfected with shPDE1A/control shRNA or PDE1A overexpressing plasmid/empty vector were suspended in 0.1 ml of PBS and injected into mice via the tail vein. After 60 days, the mice were sacrificed, and the lung tissues were collected to observe pulmonary nodules. The lung, liver, kidney, pancreas, and other tissues were separated, fixed in 4% paraformaldehyde, and stained using H&E. All animal procedures were conducted in accordance with the guidelines and regulations approved by the Institutional Animal Care and Use Committee (IACUC) of Zhejiang University City College. Ethical approval for the study was obtained under protocol number 22001.

## mRNA stability assay

NSCLC cells were seeded in six-well plates and grown to approximately 30% confluence, followed by siRNA transfection and incubation for 24 hr. Then, cells were incubated with actinomycin D (5 μg/ml) for 0, 3, or 6 hr followed by RNA extraction. The half-life of mRNA was analyzed by qRT-PCR. The mRNA expression for each group at the indicated time was calculated and normalized to GAPDH.

## Statistical analysis

Data are presented as the mean ±SD from three independent experiments. Two-tailed Student's t test was used to compare two groups. p-Values<0.05 were considered significant. *p<0.05; **p<0.01; ***p<0.001.

## Acknowledgements

It was funded by Huadong Medicine Joint Funds of the Zhejiang Provincial Natural Science Foundation of China (LHDMY22H160001), Natural Science Foundation of Ningbo City (2022J206), National Natural Science Foundation of China (82273352), the Youth Incubation Project of Xian Health Commission (2024qn01), Scientific Research Foundation of Hangzhou City College (X-202305), National College Student Innovation and Entrepreneurship Project (202313021036), and Zhejiang Provincial Medical and Health Technology Project (2025KY391).

## Additional information

### Funding

| Funder | Grant reference number | Author |
|---|---|---|
| Huadong Medicine JointFunds of the Zhejiang Provincial Natural Science Foundation of China | LHDMY22H160001 | Chong Zhang |
| Natural Science Foundation of Ningbo City | 2022J206 | Zuoyan Zhang |
| National Natural Science Foundation of China | 82273352 | Yangling Li |
| Xian Health Commission | Youth Incubation Project 2024qn01 | Zuoyan Zhang |
| Scientific Research Foundation of Hangzhou City College | X-202305 | Chong Zhang |
| National College Student Innovation and Entrepreneurship Project | 202313021036 | Zuoyan Zhang |
| Zhejiang Provincial Medical and Health Technology Project | 2025KY391 | Jian Wang |

The funders had no role in study design, data collection and interpretation, or the decision to submit the work for publication.

### Author contributions

Chong Zhang, Conceptualization, Writing – original draft, Writing – review and editing; Zuoyan Zhang, Yueyi Wu, Yuchen Wu, Jing Cheng, Kaizhi Luo, Zhidi Li, Manman Zhang, Data curation, Formal analysis; Jian Wang, Funding acquisition; Xuesen Zhang, Formal analysis; Yangling Li, Conceptualization, Supervision, Funding acquisition

### Author ORCIDs

Chong Zhang  https://orcid.org/0000-0002-8113-5585
Zuoyan Zhang  https://orcid.org/0000-0003-0883-6016
Yueyi Wu  https://orcid.org/0009-0008-3717-3884
Zhidi Li  https://orcid.org/0000-0003-0225-2028
Manman Zhang  https://orcid.org/0009-0002-0235-143X
Jian Wang  https://orcid.org/0009-0005-0457-9325
Xuesen Zhang  https://orcid.org/0009-0007-7758-252X
Yangling Li  https://orcid.org/0000-0002-4461-478X

## Ethics

The experiments were performed in compliance with the National Institutes of Health Guide for the Care and Use of Laboratory Animals. All procedures were approved by the Institutional Animal Care and Use Committee of Zhejiang University City College (No. 22001).

Reviewer #1 (Public review): https://doi.org/10.7554/eLife.98903.4.sa1
Reviewer #2 (Public review): https://doi.org/10.7554/eLife.98903.4.sa2
Author response https://doi.org/10.7554/eLife.98903.4.sa3

---

# Additional files

## Supplementary files

Supplementary file 1. The siRNA sequence used for knocking down the indicated genes.

Supplementary file 2. Primers sequences for detecting the expression of the indicated genes.

Supplementary file 3. Identification of PDE1A protein interactions by mass spectrometry.

Supplementary file 4. The potential target genes of YTHDF2 in non-small cell lung cancer (NSCLC) cells.

Supplementary file 5. The potential target genes of YTHDF2 in non-small cell lung cancer (NSCLC) cells.

MDAR checklist

## Data availability

All data generated or analysed during this study are included in the manuscript and supporting files.

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
