## [Editor Report · eLife Assessment]

This manuscript provides **valuable** mechanistic insight into NSCLC progression, both in terms of tumour metastasis and the development of chemoresistance. The authors draw upon a range of techniques and assays and the evidence shown is **solid** and has been strengthened by incorporation of suggestions by the two reviewers. The work presented will be of interest to cancer biologists and more broadly to those interested in NSCLC translational studies.

---

## [Referee Report · Reviewer #1 (Public review)]

Summary:

The manuscript entitled "Phosphodiesterase 1A Physically Interacts with YTHDF2 and Reinforces the Progression of Non-Small Cell Lung Cancer" explores the role of PDE1A in promoting NSCLC progression by binding to the m6A reader YTHDF2 and regulating the mRNA stability of several novel target genes, consequently activating the STAT3 pathway and leading to metastasis and drug resistance.

Strengths:

The study addresses a novel mechanism involving PDE1A and YTHDF2 interaction in NSCLC, contributing to our understanding of cancer progression.

---

## [Referee Report · Reviewer #2 (Public review)]

Summary:

This revised manuscript investigates the role and the mechanism by which PDE1 impacts NSCLC progression, providing solid data to demonstrate that PDE1 binds to m6A reader YTHDF2, in turn, regulating STAT3 signaling pathway through its interaction, promoting metastasis and angiogenesis. The study provides a valuable information to lung cancer field.

Strength:

The study uncovers a novel PDE1A/YTHDF2/SOCS2/STAT3 pathway in NSCLC progression and the findings provide a potential treatment strategy for NSCLC patients with metastasis.

Weakness:

Given that physical interaction of PDE1A and YTHDF2 plays a critical role in PDE1A-mediated NSCLC metastasis, the in vivo data to show that YTHDF2 mimics the effect of PDE1A in metastasis will strength the manuscript although this point was mentioned in the revised manuscript.

---

## [Author Response]

The following is the authors’ response to the previous reviews

**Public Reviews:**

**Reviewer #1 (Public review):**
Summary:The manuscript entitled "Phosphodiesterase 1A Physically Interacts with YTHDF2 and Reinforces the Progression of Non-Small Cell Lung Cancer" explores the role of PDE1A in promoting NSCLC progression by binding to the m6A reader YTHDF2 and regulating the mRNA stability of several novel target genes, consequently activating the STAT3 pathway and leading to metastasis and drug resistance.Strengths:The study addresses a novel mechanism involving PDE1A and YTHDF2 interaction in NSCLC, contributing to our understanding of cancer progression.
**Reviewer #2 (Public review):**
SummaryThis revised manuscript investigates the role and the mechanism by which PDE1 impacts NSCLC progression. They provide evidence to demonstrate that PDE1 binds to m6A reader YTHDF2, in turn, regulating STAT3 signaling pathway through its interaction, promoting metastasis and angiogenesis.Strength:The study uncovers a novel PDE1A/YTHDF2/SOCS2/STAT3 pathway in NSCLC progression and the findings provide a potential treatment strategy for NSCLC patients with metastasis.Weakness:In discussion, it is stated in the revised version that "the role of YTHDF2 in PDE1A-driven tumor metastasis should be elucidated in future studies", however, given that physical interaction of PDE1A and YTHDF2 plays a critical role in PDE1A-mediated NSCLC metastasis, whether YTHDF2 mimicking the effect of PDE1A in metastasis will strength the manuscript.
**Recommendations for the authors:**

**Reviewer #2 (Recommendations for the authors):**
(1) In Figure 1A, the y-axis should be "IOD/Area" instead of "IDO/Area".

Figure 1A was revised as suggested.

(2) Figure 3A legend for (F) and (G) was switched.

Figure 3A legend was revised as suggested “(F-G) The mRNA (F) and protein (G) levels of indicated genes were determined in P3 and P0 NSCLC cells.”.

(3) The statistical analysis should be performed for Figure 3H.

Figure 3H was revised as suggested.

(4) Figure 4F, Y-axis has a typo for "vessels" and statistical analysis should be performed on this data.

Figure 4F was revised as suggested.

(5) Figure 6 E, typo for "migrated" on the y-axis.

Figure 6E was revised as suggested.

(6) Figure 7 C, typos for "expression" on y-aixs in both figures need to be fixed.

Figure 7C was revised as suggested.

(7) P-values for Figure 7B need to be stated.

Figure 7B was revised as suggested.

(8) m6A should be consistent throughout the manuscript.

m6A was consistent throughout the manuscript.